# Knockdown of 15-bp Deletion-Type v-raf Murine Sarcoma Viral Oncogene Homolog B1 mRNA in Pancreatic Ductal Adenocarcinoma Cells Repressed Cell Growth In Vitro and Tumor Volume In Vivo

**DOI:** 10.3390/cancers14133162

**Published:** 2022-06-28

**Authors:** Jiaxuan Song, Yoshiaki Kobayashi, Yoshimasa Asano, Atsushi Sato, Hiroaki Taniguchi, Kumiko Ui-Tei

**Affiliations:** 1Department of Computational Biology and Medical Sciences, Graduate School of Frontier Sciences, The University of Tokyo, Chiba 277-8561, Japan; song_jiaxuan_18@stu-cbms.k.u-tokyo.ac.jp; 2Department of Biological Sciences, Graduate School of Science, The University of Tokyo, Tokyo 113-0033, Japan; yoshiaki-kobayashi@g.ecc.u-tokyo.ac.jp (Y.K.); yoshimasa.asano@bs.s.u-tokyo.ac.jp (Y.A.); atsushisato0728@gmail.com (A.S.); 3Keio Cancer Center, Keio University School of Medicine, Tokyo 160-8582, Japan; h-tani@keio.jp

**Keywords:** BRAF, pancreatic ductal adenocarcinoma, cancer, deletion mutation, siRNA, nucleic acid therapy

## Abstract

**Simple Summary:**

The *v-raf murine sarcoma viral oncogene homolog B1* (*BRAF)* gene containing a 15-base pair (bp) deletion at L485-P490 is the cause of several cancers. We generated siRNA to specifically knock down *BRAF* mRNA containing the 15-bp deletion. This siRNA suppressed the expression of *BRAF,* harboring the deletion without affecting wild-type *BRAF* expression in BxPC-3 pancreatic ductal adenocarcinoma cells in vitro and in vivo. Cell growth and phosphorylation of downstream extracellular-signal-regulated kinase proteins were also repressed. An off-target effect is the most common side effect of siRNA therapy. In this study, we reveal that siRNA with a 2′-O-methyl chemical modification in the seed region of the siRNA guide strand reduced seed-dependent off-target effects.

**Abstract:**

Pancreatic ductal adenocarcinoma (PDAC) is predicted to become the second-most common cause of death within the next 10 years. Due to the limited efficacy of available therapies, the survival rate of PDAC patients is very low. Oncogenic BRAF mutations are one of the major causes of PDAC, specifically the missense V600E and L485–P490 15-bp deletion mutations. Drugs targeting the V600E mutation have already been approved by the United States Food and Drug Administration. However, a drug targeting the deletion mutation at L485–P490 of the BRAF gene has not been developed to date. The BxPC-3 cell line is a PDAC-derived cell line harboring wild-type KRAS and L485–P490 deleted BRAF genes. These cells are heterozygous for BRAF, harboring both wild-type BRAF and BRAF with the 15-bp deletion. In this study, siRNA was designed for the targeted knockdown of 15-bp deletion-type BRAF mRNA. This siRNA repressed the phosphorylation of extracellular-signal-regulated kinase proteins downstream of BRAF and suppressed cell growth in vitro and in vivo. Furthermore, siRNAs with 2′-O-methyl modifications at positions 2–5 reduce the seed-dependent off-target effects, as confirmed by reporter and microarray analyses. Thus, such siRNA is a promising candidate therapy for 15-bp deletion-type BRAF-induced tumorigenesis.

## 1. Introduction

Pancreatic ductal adenocarcinoma (PDAC) is the most common type of pancreatic adenocarcinoma, accounting for up to 90% of all pancreatic malignancies [1]. To date, PDAC is the fourth leading cause of cancer-related death worldwide, with a 5-year survival rate of less than 8% [2]. The incidence of PDAC is predicted to increase twofold within the next 10 years, and PDAC is expected to become the second most common cause of cancer death [3]. A mutation of the oncogene *Kirsten rat sarcoma* (*KRAS*) is the most common cause of pancreatic cancer, accounting for more than 90% of cases [4]. The *KRAS* gene encodes a small GTPase transductor protein that binds to guanosine diphosphate (GDP) and guanosine triphosphate (GTP) [4]. KRAS possesses intrinsic GTPase activity, is involved in the Ras/mitogen-activated protein kinase (MAPK) signaling pathway, and plays an important role in cell proliferation [5]. Most KRAS mutations impair the intrinsic activity of GTPase, impede the function of GTPase-accelerating proteins, and inhibit the conversion of GTP to GDP [5]. KRAS is thus permanently bound to GTP, activating downstream signaling pathways including cell proliferation [6,7]. In addition to PDAC harboring KRAS mutations, 10% of PDAC cases harbor wild-type (WT) KRAS [1]. Among these cases, 40% are related to defects in the activated MAPK pathway caused by oncogenic BRAF mutation, and 20% are due to defective DNA mismatch repair resulting from microsatellite instability in tumors [1]. The remaining 40% of cases are due to kinase fusion genes [1]. Among PDAC cases with BRAF mutations, the most common mutation (70–88% of cases) is V600E [8]. The V600E point mutation promotes cell proliferation via the abnormal activation of the Ras/MAPK signaling pathway [9]. To date, two drugs for the V600E mutation have been approved by the United States Food and Drug Administration (FDA): vemurafenib in 2011 [10] and dabrafenib in 2013 [11]. Aside from V600E, the rate of *BRAF* in-frame deletions with KRAS WT pancreatic cancer is shown to be 4.21% [11]. Among them, 15-base pair (bp) deletion from L485 to P490 in BRAF, an activating mutation, has also been detected [11]. Structurally, the region of BRAF containing 15-bp deletion is involved in the β3/αC-helix loop, and this deletion shortens the β3/αC homologous loop within the kinase domain [12]. Therefore, the 15-bp deletion reduces flexibility by locking the helix into the active αC-helix conformation, generating a constitutively active form [11]. This 15-bp in-frame deletion has been observed in lung and ovarian cancers in addition to pancreatic cancers [11]. On the other hand, the BRAF knockout mouse is embryonic lethal due to an increased number of endothelial precursor cells and apoptotic death of differentiated endothelial cells [13]. Therefore, the technology to specifically suppress the *BRAF* mutation (Mut) without affecting the WT *BRAF* expression is necessary for therapeutics. In this study, we used PDAC-derived BxPC-3 cells, which are one of the first identified cancer-derived cell lines, and the 15-bp deletion Mut of *BRAF* is shown to be a cause of carcinogenesis [11].

RNA interference (RNAi) is a widely used gene silencing mechanism that can specifically knock down a target gene with sequence complementarity [14,15]. In RNAi, small interfering RNA (siRNA) acts as a mediator of mRNA degradation [16,17,18,19,20,21,22]. In this study, synthetic siRNA, a duplex composed of the guide and passenger strands of 21-nucleotide-long RNAs with 2-nucleotide 3′ overhangs, was used. The siRNAs transfected into the cells are taken up by the RNA-induced silencing complex and loaded onto the catalytic subunit of the Argonaute2 (AGO2) protein. On AGO2, the siRNA is unwound into single-stranded RNAs; the guide strand, which has a less stable 5′ end compared with its 3′ end, typically remains on AGO2, whereas the other strand is degraded [23,24,25,26,27]. The siRNA guide strand undergoes base pairing with complementary target mRNA and cleaves it via the catalytic activity of AGO2, thus silencing the target gene.

Although siRNA is a useful tool for silencing target genes, extensive evidence indicates that the specificity of siRNA is not absolute [28]. Some off-target effects of RNAi have been reported [29,30]. Among them, the most frequently occurring off-target effect is the siRNA seed-dependent effect, which is induced by the partial complementarity of the siRNA seed region (positions from 2–8 from the 5′ end) to the sequences in the 3′ untranslated regions (UTRs) of unexpected off-target genes [31,32]. This effect is strongly correlated with the thermodynamic stability of the duplex formed between the seed region of the siRNA guide strand and its off-target mRNA [32,33], with higher seed-target stability leading to greater off-target effects, which are sequence-dependent. Although siRNAs have off-target effects, several processes can mitigate these effects [15]. Various chemical modifications have been developed to modify the biochemical properties of siRNA [34]. The 2′-O-methy (2′-OMe) modification is a common nucleoside modification of RNA, which involves adding a methyl group to the 2′ hydroxyl group of the nucleoside [35]. In our previous reports, we revealed that 2′-OMe modifications in the seed region of siRNA significantly reduced off-target effects via the induction of steric hindrance in the duplex between the siRNA guide strand and mRNA on the AGO protein without affecting RNAi activity [36].

In this study, we established an siRNA that specifically suppresses expression of the Mut *BRAF* gene and exhibits negligible off-target effects on the expression of WT *BRAF*. We demonstrated that this siRNA clearly reduced the expression of Mut BRAF without affecting the expression of WT BRAF and also suppressed the phosphorylation of extracellular-signal-regulated kinase (ERK)1 and 2 proteins in the downstream signaling pathway that potentially inhibits cell growth. This siRNA was also effective in a xenograft tumor mouse model.

## 2. Materials and Methods

### 2.1. Chemical Synthesis of siRNAs

Both the unmodified and modified siRNAs were chemically synthesized (Shanghai GenePharma, and Gene Design). All siRNA sequences are shown in Appendix A.

### 2.2. Cell Culture

The human PDAC-derived cell lines, BxPC-3 (ATCC CRL-1687) and AsPC-1 (ATCC CRL-1682), were cultured at 37 °C with 5% CO_2_ in RPMI 1640 medium (Thermo Fisher Scientific, Waltham, MA, USA) with 10% heat-inactivated fetal bovine serum (FBS, Sigma, St. Louis, MO, USA) and 1% penicillin-streptomycin solution (FUJIFILM Wako, Richmond, VA, USA), essentially according to the product sheet of American Type Culture Collection (ATCC). Human HeLa cells, derived from cervical cancer, and OCUB-M cells (RCB0881), derived from breast cancer, were cultured at 37 °C with 5% CO_2_ in Dulbecco’s Modified Eagle’s medium (DMEM, Thermo Fisher Scientific) with 10% heat-inactivated FBS and 1% penicillin-streptomycin solution, essentially according to the cell culture procedure of RIKEN BRC.

### 2.3. Construction of Plasmids for Complete-Matched (CM) and Seed-Matched (SM) Luciferase Reporter Assays

All the reporter plasmids were constructed from the psiCHECK-1 vector (Promega, Madison, WI, USA). Oligonucleotides used for insertion into the psiCHECK-1 vector were synthesized with the *Xho*I or *EcoR*I sticky end at each end. Then, the synthesized oligonucleotides were inserted into the corresponding restriction enzyme sites, which are located at the 3′ UTR region of *Renilla luciferase* gene in psiCHECK-1 vector. The plasmids that contained CM sequences were synthesized for testing the siRNA effect. The plasmids that contained three tandem repeats of SM sequences were synthesized for testing the siRNA off-target effect. The sequences of the inserted oligonucleotides are shown in Appendix A.

### 2.4. Measurements of RNAi and Off-Target Activities by Dual Luciferase Reporter Assays

Since three types of cells, including HeLa, AsPC-1, and OCUB-1 cells, have been used for evaluating siRNA activities [36,37,38], these cells were selected for the luciferase reporter assay. Cells were inoculated in a well of 24-well culture plates (1 × 10^5^ cells/well), respectively. Twenty-four hours later, cells were transfected with siRNA (final concentration of 0.005, 0.05, 0.5, or 5 nM), 100 ng of pGL3-Control vector (Promega), and 10 ng of the corresponding psiCHECK-1 vector using Lipofectamine 2000 reagent (Thermo Fisher Scientific) according to the manufacture’s protocol. pGL3-Control vector encodes the firefly *luciferase* gene, which was used as an internal control of luciferase activity. Control siRNA, siControl, does not target either CM- or SM-reporter constructs [31,36]. At 24 h after transfection, cells were lysed by 1 × passive lysis buffer (Promega). Luciferase activity was measured using the Dual-Luciferase Reporter Assay System (Promega) and GloMax Discover Microplate Reader (Promega). The RNAi or off-target activity by the transfection of each siRNA was calculated by *Renilla* luciferase activity normalized by firefly luciferase activity, and the relative percentage was calculated compared to the result of the siControl.
$$RNAi\;or\;off-target\;activity=\frac{Renilla\;luciferase\;activity}{Firefly\;luciferase\;activity}$$

### 2.5. Quantitative Reverse Transcription-Polymerase Chain Reaction

BxPC-3 cells have no mutated KRAS genes, but they have L485–P490 deleted *BRAF* genes [11], and AsPC-1 cells have mutated KRAS genes [39]. Therefore, we used these two cell lines. BxPC-3 and AsPC-1 cell suspensions (0.5 × 10^5^ cells/well) were seeded into each well of 24-well culture plate 24 h before transfection, respectively. Cells were transfected with 50 nM siRNA by Lipofectamine 2000 or Lipofectamine RNAiMAX reagent (Thermo Fisher Scientific) on 3 consecutive days (see Figure 1A and Figure 2B). At 24 or 48 h after the third siRNA transfection, the cells were collected. Total RNAs were extracted from the cells using the FastGene RNA Premium Kit (Nippon Genetics) according to the manufacture’s protocol. The RNA extracted from the siControl-treated cells was used as a control. Each 0.3 μg of the total RNA was used for cDNA synthesis with the High-Capacity cDNA Reverse Transcription Kit (Applied Biosystems). qRT-PCR was performed by two-step cycle method (Hold; 95 °C, 10 min, PCR (40×); 95 °C, 15 s, and 60 °C, 1 min) using the KAPA SYBR Fast qPCR Kit (KAPA BIOSYSTEMS) with the StepOnePlus Real-Time PCR System (Applied Biosystems) and analyzed by the ΔΔCt method. The expression level of each sample was normalized by the expression level of glyceraldehyde-3-phosphate dehydrogenase (GAPDH). The primer sequences are shown in Appendix A.

### 2.6. Western Blotting

For western blotting, the cells were scraped off from the dish using a plastic cell scraper. The cells were washed with PBS and lysed in lysis buffer (10 mM Hepes-NaOH (pH 7.9), 1.5 mM MgCl_2_, 10 mM KCl, 0.5 mM dithiothreitol (DTT) (Nacalai Tasque), 140 mM NaCl, 1 mM EDTA, 1 mM Na_3_VO_4_, 10 mM NaF, 0.5% nonidet p-40 (NP-40) (FUJIFILM Wako), and complete protease inhibitor (Merck)). After determining the protein concentration and denaturization, the samples were separated by sodium dodecyl sulfate polyacrylamide gel electrophoresis and transferred to an Immobilon-P polyvinylidene fluoride membrane (Merck Millipore) using the Mini Trans-Blot Cell (Bio-Rad). The membrane was blocked for 1 h in Tris-buffered saline with Tween 20 (TBS-T; 20 mM Tris–HCl [pH 7.5], 150 mM NaCl, 0.1% Tween) supplemented with 5% skim milk (FUJIFILM Wako) at room temperature and incubated with specific first antibodies in TBS-T with 5% skim milk at 4 °C overnight. The membranes were washed three times with TBS-T and reacted with the second antibodies at room temperature for 1 h. After being washed three times with TBS-T, the membrane was incubated with ECL Prime Western Blotting Detection Reagent (Cytiva) and visualized using the FUSION (VILBER). The first antibodies used are as follows: rabbit anti-phospho-ERK1/2 antibody at 1: 1000 dilution (Cell Signaling Technology, 4370), rabbit anti-ERK1/2 antibody at 1:1000 dilution (Cell Signaling Technology, 4695), and mouse anti-β-actin antibody at 1: 4000 dilution (Merck, A2228). The second antibodies used are as follows: horseradish peroxidase (HRP)-linked anti-rabbit or anti-mouse antibody at 1:3000 dilution (Cytiva, NA934 or NA931). The protein level of each sample was measured using software of FUSION and normalized by that of β-actin.

### 2.7. Microarray

The cell suspension (1.0 × 10^5^ cells/mL) of HeLa cells was inoculated into a well of a 24-well culture plate 1 day before transfection. Cells were transfected three times with 50 nM of each siRNA using Lipofectamine 2000 according to the manufacture’s protocol, since most of the cells were transfected with siRNAs by three transfections. The total RNA of the harvested HeLa cells was purified using the FastGene RNA Premium Kit (NIPPON Genetics, Tokyo, Japan) according to the manufacture’s protocol, and the RNA quality was confirmed using a NanoDrop 2000 spectrophotometer (Thermo Fisher Scientific) and Bioanalyzer (Agilent Technologies, Santa Clara, CA, USA). cDNA and Cy3-labeled RNA were synthesized using the Quick Amp Labeling Kit for one color (Agilent Technologies). Cy3-labeled RNAs were fragmented with the Gene Expression Hybridization Kit (Agilent Technologies) and hybridized to a SurePrint G3 Human GE Microarray (version 3.0, 8 × 60 K) (Agilent Technologies) at 65 °C for 17 h. After washing, the microarray slide was scanned by a DNA Microarray Scanner (Agilent Technologies), and the signals were quantified by Feature Extraction 10.5.1.1 software (Agilent Technologies). RNA from mock-transfected cells treated with the transfection reagent in the absence of siRNA was used as a control, and the distributions of the signal intensities of transcripts were normalized across all samples by quantile normalization [40]. Results were shown in MA plots and cumulative accumulations. In the MA plot, M indicates the intensity ratio and A indicates the average intensity. Furthermore, the aliguot of total RNA was used for qRT-PCR for the validation of microarray data. The expression level of each sample was normalized by GAPDH expression level. The primer sequences are shown in Appendix A.

### 2.8. Xenograft Mouse Model

NOG mice (NOD.Cg-*Prkdc*^scid^*Il2rg*^tm1Sug^/ShiJic, 6 weeks old, male) were obtained from CLEA Japan (Japan) and individually housed in cages with free access to food and water and reared in 12-h light–dark cycles in a light-tight chamber at 24 ± 1 °C. BxPC-3 cells (2.0 × 10^6^ cells) in HBSS (SIGMA, Japan) were implanted into the lateral region of the abdomen of the mice subcutaneously. At four days after the tumor implantation, each siRNA was injected into a BxPC-3 cell implant intratumorally using in vivo jetPEI (Polyplus-transfection, France) every week. The tumor size was measured by caliper twice a week, and the tumor volume was calculated using the following formula (modified Hansen formula): V = L × S^2^ × 1/2 (L: longer diameter of tumor, S: shorter diameter of tumor). This formula is widely accepted for estimating the xenograft tumor volume [41,42]. All procedures involving animals were approved by the Animal Ethics Committee of the University of Tokyo (Animal Plan ID: 31–13).

### 2.9. Serum Degradation Assay

Serum degradation assays were performed by incubating 1 µL of 20 µM siRNA in 10 µL of PBS solution containing 10% heat-inactivated FBS (Sigma) at 37 °C. After treatment for the desired duration, the solution was immediately frozen in liquid nitrogen and stored at −80 °C until gel electrophoresis, in which 2 µL samples were separated in 20% polyacrylamide gels and visualized by ethidium bromide staining.

## 3. Results

### 3.1. Effect of Kirsten Rat Sarcoma Viral Oncogene Homolog Knockdown in PDAC-Derived Cell Lines AsPC-1 and BxPC-3, with or without Mut Kirsten Rat Sarcoma Viral Oncogene Homolog

More than 90% of patients with pancreatic cancers have KRAS mutations, and BRAF deletion mutations are mutually exclusive with KRAS mutations [11]. Therefore, at first, the effect of siRNA targeting KRAS (siKRAS) (Appendix A) was examined in two different pancreatic cancer-derived cell lines, AsPC-1 and BxPC-3. AsPC-1 cells harbor homozygous KRAS mutations in the GTP binding sites (35G > A) [39], whereas BxPC-3 cells harbor WT KRAS gene and Mut BRAF (15-bp deletion at L485-P490) [11]. The siKRAS used in this study repressed both WT and 35G > A-mutated KRAS gene expressions simultaneously because the common region was used as a target site. After transfection of siKRAS for 3 consecutive days, the cells were harvested 1 day after the last transfection, and qRT-PCR was performed to quantify the total KRAS mRNA (Figure 1A). Relative to the control cells transfected with the control siRNA (siControl), the KRAS mRNA level effectively decreased to 21.2% in siKRAS-treated AsPC-1 cells (Figure 1B). One of the major downstream effects of KRAS activation is phosphorylation of the rapidly accelerated fibrosarcoma mitogen-activated protein kinase (MAPK) kinases ERK1/2 [11,43]. Therefore, to investigate the effect of siKRAS on this downstream phosphorylation, western blotting was performed to quantify the phosphorylation levels of ERK1 and ERK2. siKRAS effectively suppressed the phosphorylation of ERK1/2 in siKRAS-transfected AsPC-1 cells (Figure 1C), suggesting that the downstream cell proliferation signaling of the cells is active and functional in response to KRAS knockdown. Therefore, the effect of siKRAS on cell growth was measured. The cell numbers relative to those on day 1 after the last siKRAS transfection were determined on days 2, 3, 4, and 5 (Figure 1D). Compared with cells treated with siControl, siKRAS significantly suppressed the number of AsPC-1 cells. The same experiments were performed in BxPC-3 cells. Although siKRAS significantly decreased the KRAS mRNA level to approximately 6% of the control level (Figure 1E), it showed negligible effects on the phosphorylation levels of ERK1/2 (Figure 1F) and the number of BxPC-1 cells (Figure 1G). Thus, the suppression of the KRAS expression by siKRAS in AsPC-3 cells successfully repressed KRAS mRNA expression, KRAS downstream signaling, and cell number, suggesting that the downstream proliferation pathways of AsPC-3 cells are active and functional in response to the KRAS knockdown. However, a negligible effect in BxPC-3 cells harboring WT KRAS was observed, despite a reduction in the KRAS mRNA level, suggesting that the function of BRAF deletion mutations is mutually exclusive with KRAS mutations.

### 3.2. Effect of Knockdown of Mut BRAF in BxPC-3 Cells

The PDAC-derived cell line BxPC-3 possesses WT KRAS, but is heterogenous for BRAF, harboring both WT and Mut BRAF. We verified BRAF heterozygosity in BxPC-3 cells by genomic sequencing. We designed three types of siRNAs targeting BRAF mRNA (Figure 2A): siBRAF_WT&Mut was designed to knock down both WT and Mut BRAF, siBRAF_WT was designed to knock down WT BRAF mRNA only, and siBRAF_Mut1 and siBRAF_Mut2 were designed to knock down Mut BRAF mRNA only (Figure 2A). Each siRNA (50 nM) was transfected into BxPC-3 cells daily on days 1–3 using Lipofectamine 2000 (Figure 2B). Three transfections were performed to transfect the siRNAs into almost all of the cells and did not show fatal damage to the cells, since the downstream responses by BxPC-3 knockdown were successfully observed. At 24 and 48 h after the third transfection, cells were collected and qRT-PCR was performed to quantify the expression of total BRAF mRNA, including both WT and Mut BRAF mRNA using common primers (Figure 2C). The transfection of siBRAF_WT&Mut decreased the total mRNA amount of BRAF to the lowest level, since it knocked down both WT and Mut types of BRAF mRNA simultaneously. At 24 h after transfection, siBRAF_WT was as effective as siBRAF_WT&Mut and sufficiently decreased the BRAF mRNA level (Figure 2D). However, the BRAF mRNA level recovered at 48 h after transfection with siBRAF_WT, which was earlier than with siBRAF_WT&Mut. Both siBRAF_Mut1 and siBRAF_Mut2 decreased BRAF mRNA expressions to approximately 50–60% of the initial level at both 24 and 48 h. These results indicate that all BRAF siRNAs showed stronger inhibitory effects at 24 h compared with 48 h, and, therefore, qRT-PCR and western blotting was performed at 24 h after three consecutive transfections.

Next, qRT-PCR was performed to quantify the expression levels of each of WT and Mut BRAF separately. Primers were designed to detect WT or Mut BRAF mRNA individually (Figure 2C). Using these primers, it was revealed that both siBRAF_WT&Mut and siBRAF_WT effectively suppressed the expression of WT BRAF mRNA (Figure 2E), and siBRAF_WT&Mut, siBRAF_Mut1, and siBRAF_Mut2 effectively suppressed the expression of Mut BRAF mRNA (Figure 2E).

We further examined the downstream signaling. Western blotting revealed nearly the same ERK1/2 phosphorylation level between siBRAF_WT and siControl transfections, whereas siBRAF_WT&Mut, siBRAF_Mut1, and siBRAF_Mut2 markedly suppressed ERK1/2 phosphorylation (Figure 2F). Then, the effects of these siRNAs on BxPC-3 cell numbers were examined. The cell numbers relative to those on day 1 after siRNA transfection were determined on days 2, 3, 4, and 5 (Figure 2G). Compared with siControl-treated cells, siBRAF_WT&Mut, siBRAF_Mut1, and siBRAF_Mut2 showed significant inhibitory effects on the number of BxPC-3 cells, whereas siBRAF_WT had no such inhibitory effect (Figure 2G). Thus, the knockdown of Mut BRAF suppressed both ERK phosphorylation and cell numbers, but WT BRAF knockdown did not suppress ERK phosphorylation and cell numbers. These results may indicate that the expression levels of WT and Mut BRAF genes themselves or the phosphorylation levels of ERK by WT and Mut BRAF are different in BxPC-3 cells.

### 3.3. Effects of v-Raf Murine Sarcoma Viral Oncogene Homolog B1 siRNAs on RNAi and off-Target Activities

To investigate the effects of siRNAs targeting WT or Mut BRAF mRNA on RNAi and off-target activities, luciferase reporter assays were performed using the strategy shown in our previous reports [31,36]. Oligonucleotides containing a CM sequence or three tandem repeats of the SM sequence were chemically synthesized, inserted into the 3′UTR of the Renilla luciferase gene in the psiCHECK-1 vector, and designated psiCHECK-CM and psiCHECK-SM, respectively (Figure 3A). As CM or SM target sequences, the complementary sequence of each siRNA guide strand shown in Figure 3A was used. psiCHECK-CM and psiCHECK-SM were individually transfected into human HeLa cells along with pGL3-Control, which expresses the firefly luciferase gene as a control, and siRNAs at various concentrations. Luciferase activity was measured 1 day after transfection. siBRAF_WT&Mut strongly suppressed Renilla luciferase activity in the cells harboring the CM target sequence in the 3′UTR of the luciferase gene, but not in the cells harboring the SM target sequence (Figure 3B), indicating that siBRAF_WT&Mut has strong RNAi activity but negligible off-target activity. siBRAF_WT suppressed the expression of both the CM and SM targets. Thus, siBRAF_WT has off-target activity in addition to RNAi activity. Both siBRAF_Mut1 and siBRAF_Mut2 reduced the expression of CM targets and also repressed the expression of SM targets, indicating that both of them had obvious off-target activities in addition to strong RNAi activities.

The same experiments were performed using another cell line, OCUB-M, derived from human breast cancer (Appendix A). The results were essentially the same as those obtained using HeLa cells. The siBRAF_WT&Mut exhibited strong RNAi activity, with negligible off-target activity. In contrast, siBRAF_WT, siBRAF_Mut1, and siBRAF_Mut2 showed obvious off-target activities along with strong RNAi activities.

### 3.4. Effects of v-Raf Murine Sarcoma Viral Oncogene Homolog B1 siRNA with 2′-OMe Modification in the Seed Region on RNAi and off-Target Activities

In our previous report [36,44,45], siRNA off-target activity was effectively reduced by introducing 2′-OMe modifications in the seed region of the siRNA guide strand, since the structural change by introduction of 2′-OMe modification induces steric hindrance in interrupting duplex formation on the AGO protein. To overcome the seed-dependent off-target effects induced by siBRAF_WT, siBRAF_Mut1, and siBRAF_Mut2, 2′-OMe modifications were introduced into the seed region (nucleotides 2–5). The 2′-OMe modified siRNAs were designated as siBRAF_WT&Mut_2′-OMe, siBRAF_WT_2′-OMe, siBRAF_Mut1_2′-OMe, and siBRAF_Mut2_2′-OMe, respectively. Then, luciferase reporter assays were performed using these modified siRNAs (Figure 3B) according to our previous report [36,44]. siBRAF_WT_2′-OMe showed a similar level of RNAi activity to that of siBRAF_WT, but its off-target activity was reduced almost completely. siBRAF_Mut1_2′-OMe and siBRAF_Mut2_2′-OMe showed similar levels of RNAi activities compared to those of siBRAF_Mut1 and siBRAF_Mut2, respectively, but their off-target activities were apparently reduced compared to those of siBRAF_Mut1 and siBRAF_Mut2. Thus, 2′-OMe modification can reduce off-target activity without affecting RNAi activity, although the degree of off-target activity reduction varied. The results of AsPC-1 (Appendix A) and OCUB-M cells (Appendix A) were almost the same as those of HeLa cells shown in Figure 3.

### 3.5. Genome-wide Analyses of siRNA Off-Target Effects by Microarray

To investigate the genome-wide off-target effects of siBRAFs without/with 2′-OMe modifications, microarray analysis was performed. Each siBRAF was transfected into HeLa cells, and total RNA was purified from cells 1 day later and subjected to microarray analyses (Figure 4). In this experiment, we used HeLa cells, which are suitable to having genome-wide data, although these cells do not have mutated a BRAF gene. In contrast, we tried to take microarray data using BxPC-3 cells harboring the 15-bp BRAF deletion mutation; these cells are very weak, and transfection damage is strong. Therefore, we could not take stable data using BxPC-3 cells. The MA plots (M = intensity ratio, A = average intensity) of the microarray data using HeLa cells (Figure 4A,C,E,G,I,K,M) indicated changes in the expression levels of the transcripts, and dark blue circles indicated the off-target transcripts with each SM sequence of nucleotides 2–8 of the guide strand in their 3´ UTRs. The cumulative distribution indicated the average fold-changes of the off-target transcripts compared to those with non-SM transcripts (Figure 4B,D,F,H,J,L,N). The expression levels of the WT BRAF gene in HeLa cells were significantly downregulated by siBRAF_WT and siBRAF_WT&Mut, but not by siBRAF_Mut1 and siBRAF-Mut2 (Figure 4O). Such clear suppression of BRAF expression was also observed by siBRAF_WT_2′-OMe at a similar level. Consistent with the results of the reporter assay, siBRAF-WT&Mut did not show an off-target effect (Figure 4A,B). Although siBRAF-WT showed a significant reduction in seed-matched off-target transcripts (*p* = 5.2 × 10^−4^, Figure 4C,D), such off-target transcripts were apparently eliminated by siBRAF_WT_2′-OMe (*p* = 0.14, Figure 4E,F), indicating that 2′-OMe modifications in the siBRAF-WT seed region reduced the off-target effect. Similarly, siBRAF-Mut1 (Figure 4G,H) and siBRAF-Mut2 (Figure 4K,L) showed a significant reduction in off-target transcripts, but their off-target effects were obviously eliminated by 2′-OMe modifications at the seed region (Figure 4I,J,M,N). The difference in the mean log_2_ fold-changes of off-target transcripts of each siRNA was calculated as an indicator of the degree of downregulations of off-target mRNAs (Figure 4P). The unmodified siBRAF_WT showed an off-target effect, whereas its off-target effect was clearly suppressed by 2′-OMe modifications in the seed region. Similarly, siBRAF_Mut1 and siBRAF-Mut2 also showed significant off-target effects, but the 2′-OMe modifications in the seed regions of both of siRNAs significantly reduced the off-target effects. These results suggest that off-target effects on the endogenous genes are clearly prevented by 2′-OMe modifications at positions 2–5. In addition, it was noted that microarray data are essentially identical to the results of qRT-PCR (Appendix A).

### 3.6. Suppression of Cell Growth of BxPC-3 Cells by siBRAF-Muts in Mice

To evaluate the anti-tumor effects of siBRAFs with 2′-OMe modifications on BxPC-3 cells in vivo, we investigated the effects of 2′-OMe-modified siBRAFs in BxPC-3 cells. At first, each siRNA (50 nM) was transfected into BxPC-3 cells daily on days 1–3 using Lipofectamine 2000. At 24 h after the third transfection, cells were collected and qRT-PCR was performed to quantify the expression of WT and Mut BRAF mRNAs, respectively, using each of the distinguishable PCR primers (Figure 2C) for WT or Mut BRAF mRNAs. All of the siBRAF siRNAs (siBRAF-WT, siBRAF-Mut1, siBRAF-Mut2) with 2′-OMe modifications showed similar levels of suppression on BRAF mRNAs compared to the non-modified ones (Appendix A). However, it is unknown whether these results reflect the in vivo conditions. Therefore, we used a subcutaneous xenograft mouse model implanted with BxPC-3 cells. Four days after the implantation, 5 µg of each siRNA was intratumorally injected once a week (Figure 5A). The growth of the tumor was suppressed by the treatment with siBRAF_Mut1_2′-OMe or siBRAF-Mut2_2′-OMe, compared to the result of siControl treatment (Figure 5B). These findings demonstrate that both siBRAF_Mut1_2′-OMe and siBRAF-Mut2_2′-OMe inhibit the proliferation of a BRAF-mutated pancreatic tumor in the xenograft mouse model.

## 4. Discussion

We first verified that the suppression of KRAS expression by siKRAS in BxPC-3 cells did not affect the phosphorylation level of ERK1/2 downstream KRAS signaling or cell growth, although siKRAS markedly reduced the *KRAS* mRNA level (Figure 1). In contrast, siKRAS suppressed the growth of AsPC-1 cells, which are homozygous for the oncogenic *KRAS* mutation. However, siRNAs against 15-bp deleted BRAF mutants (siBRAF-Muts) reduced the mRNA levels of 15-bp deleted BRAF, downstream phosphorylation signals, and cell growth in BxPC-3 cells. Therefore, the results suggest that the 15-bp deletion of the *BRAF* mutation functions as the main cause of carcinogenesis, independent of KRAS expression in BxPC-1 cells. Although all the siRNAs targeting WT and Mut *BRAF* used in this study repressed the expression of the corresponding *BRAF* mRNAs, only those targeting Mut *BRAF* (siBRAF_WT&Mut, siBRAF_Mut1, siBRAF_Mut2) showed predominant suppressive effects on ERK1/2 phosphorylation and cell growth in BxPC-3 cells (Figure 2 and Figure 3). As the siRNA targeting WT *BRAF* (siBRAF_WT) did not suppress ERK1/2 phosphorylation or cell growth, the approximately 70% reduction in WT *BRAF* mRNA expression appears to have a negligible effect on cell growth (Figure 2E,G). In contrast, the approximately 70% reduction in Mut *BRAF* mRNA might be sufficient to suppress downstream carcinogenic signaling. These results suggest that the functions of WT and Mut *BRAF* mRNAs are different, or the absolute expression level of each mRNA is different.

All tested siRNAs targeting *BRAF* showed strong RNAi effects on their corresponding targets (Figure 2 and Figure 3, Appendix A). Furthermore, siBRAF_Mut&WT and siBRAF_Mut1 showed negligible off-target effects, whereas siBRAF_WT, siBRAF_Mut1, and siBRAF_Mut2 exhibited strong off-target effects. In this study, we examined the effects of 2′-OMe modifications at positions 2–5 of the siRNA guide strand. The seed-dependent off-target effects of siBRAF_WT_2′-OMe were almost clearly eliminated, and those of siBRAF_Mut1_2′-OMe and siBRAF_Mut2_2′-OMe were significantly reduced (Figure 3, Appendix A). As a different type of off-target effect, the innate immune response is often activated by siRNA [47]. Interferon-β (IFN-β) and 2′-5′-oligoadenylate synthetase 1 (OAS-1) are known to be upregulated during interferon response. Therefore, we investigated whether IFN-β and OAS-1 are upregulated by siRNA transfection. The results showed that the unmodified siRNAs, siBRAF-Mut1 and siBRAF-Mut2, induced IFN-β expression, and siBRAF-Mut2 also induced OAS-1. However, all of the 2′-OMe-modified siRNAs, siBRAF-WT_2′-OMe, siBRAF-Mut1_2′-OMe, and siBRAF-Mut2_2′-OMe, upregulated neither IFN-β nor OAS-1 (Appendix A), suggesting that modified siRNAs suppress innate immune response compared to the unmodified ones. Furthermore, 2′-OMe-modified siBRAF-Mut1_2′-OMe and siBRAF-Mut2_2′-OMe exhibited strong suppression activities on Mut BRAF mRNA in both in vitro (Appendix A) and in vivo xenograft tumor models (Figure 5). Thus, we successfully generated 15-bp deletion-type BRAF-specific siRNAs with almost no seed-dependent off-target effect as well as innate immune response. The knockdown of Mut BRAF using these siRNAs showed that the 15-bp deletion is the main cause of BxPC-3 carcinogenesis via the activation of downstream signaling pathways, which leads to abnormal cell proliferation (Figure 2). However, in the in vivo xenograft model, the tumors had grown slightly even after the administration of siBRAF-Muts. For the slight growth of tumors in vivo, we assume two possible reasons: (1) It might be difficult to deliver siRNAs to the cells in the deep position of the tumor cell mass when the tumor cells injected into the mice grow up to a three-dimensionally large cell mass. (2) The tumors might acquire resistance to the used siRNAs against BRAF mutations. In the case of BRAF^V600E^, which is the most frequent mutation in BRAF, it is reported that the MAPK or alternative escape pathway was reactivated during the treatment of BRAF^V600E^ inhibitors [43,48,49]. Therefore, the escaped survival pathway might also be activated in the case of siRNAs against the 15-bp deleted BRAF mutation. In such cases, the combinatorial treatment of siBRAF with MEK or PI3K signaling inhibitors may effectively reduce tumor growth.

In 2018, the first siRNA drug, patisiran (Onpattro) was approved by the FDA [45]. Even though siRNA is a powerful tool for gene silencing, many challenges and barriers to siRNA therapeutics remain to be overcome [50,51]. This is the first report of the establishment of specific siRNAs against 15-bp deletion-type Mut *BRAF* with little or no effect on the expression of the WT *BRAF* gene. However, the delivery of RNA drugs into the correct tissues, such as the pancreas, lung, or ovary, remains challenging [52]. Furthermore, establishing stable siRNAs is essential, as the naked form of RNA is easily degraded by RNases in the human body [53], although 2′-OMe-modified siRNAs (siBRAF-WT_2′-OMe, siBRAF-Mut1-2′-OMe, and siBRAF-Mut2-2′-OMe) were slightly stable compared to unmodified naked siRNAs in the serum (Appendix A). Despite these challenges and barriers, siRNA is a promising gene therapy. No drug has yet been developed for this type of *BRAF* mutation, and, therefore, the siRNAs described here are promising candidates for siRNA therapy.

## Figures and Tables

**Figure 1 cancers-14-03162-f001:**
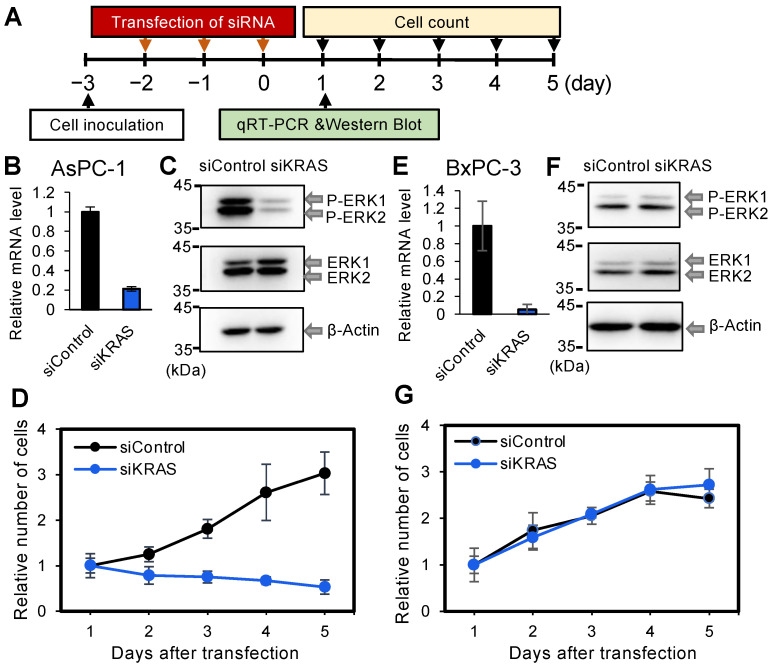
Effects of siKRAS on the endogenous KRAS mRNA expression, KRAS signaling, and numbers of AsPC-1 and BxPC-3 cells. (**A**) Experimental procedure used to investigate the effects of siKRAS on AsPC-1 and BxPC-3 cells. (**B**) KRAS mRNA levels quantified by qRT-PCR in AsPC-1 cells on day 1 after three transfections of siControl or siKRAS at 50 nM. The result for siControl was set to 1, and KRAS mRNA level relative to siControl is shown. The results indicate the averages of 3 independent experiments. (**C**) Western blotting using lysates of AsPC-1 cells transfected with siControl or siKRAS. Unphosphorylated ERK1/2 (ERK1/2) and phosphorylated ERK1/2 (P-ERK1/2) were detected by anti-ERK1/2 and anti-phospho-ERK1/2 antibodies, respectively; β-actin was used as a control. (**D**) Number of AsPC-1 cells relative to those on day 1 after three transfections of siControl or siKRAS. The black line indicates the number of AsPC-1 cells transfected with siControl, and the blue line indicates the number of cells transfected with siKRAS. The results indicate the averages of 3 independent experiments. (**E**) KRAS mRNA levels quantified by qRT-PCR in BxPC-3 cells on day 1 after three transfections of siControl or siKRAS transfections at 50 nM. The results indicate the averages of 3 independent experiments. (**F**) Western blotting using lysates of BxPC-3 cells transfected with siControl or siKRAS. (**G**) Relative number of BxPC-3 cells from 1 day after three transfections of siControl or siKRAS. The black line indicates the number of BxPC-3 cells transfected with siControl, and the blue line indicates the number of cells transfected with siKRAS. The results indicate the averages of 3 independent experiments.

**Figure 2 cancers-14-03162-f002:**
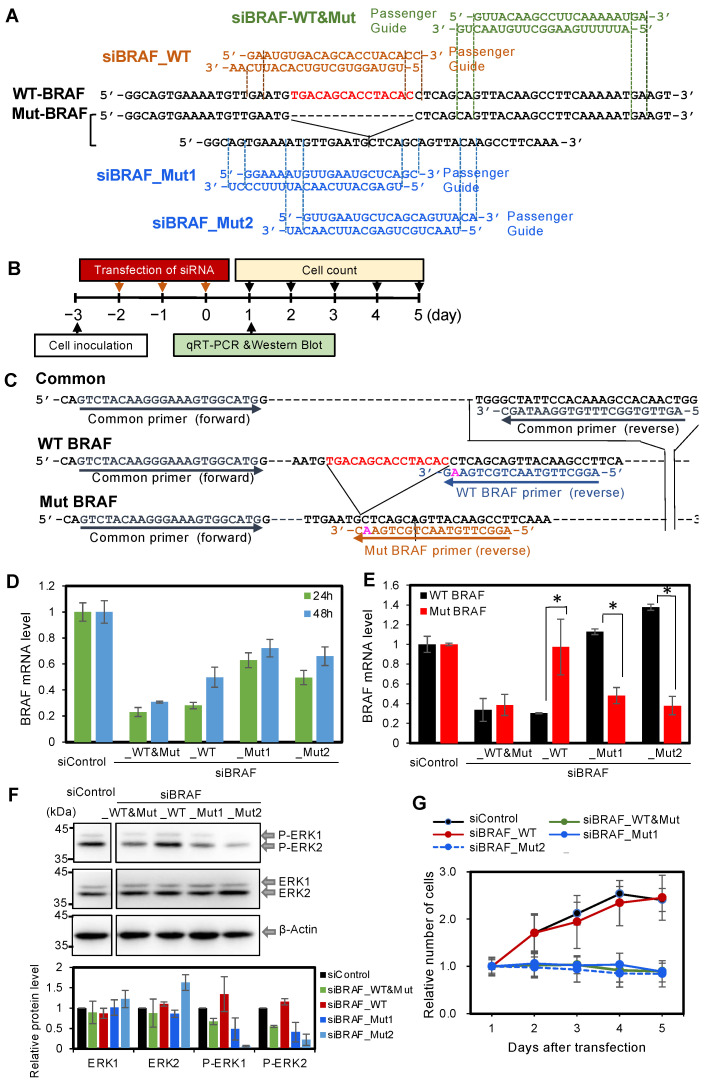
Specific siRNAs targeting WT and Mut BRAF mRNAs and their effects. (**A**) Heterologous BxPC-3 cells with WT and Mut BRAF genes. The red region indicates the 15-bp deletion in Mut BRAF. siBRAF_WT (brown) was designed to target WT BRAF mRNA only, and siBRAF_Mut1 and siBRAF_Mut2 (blue) were designed to target Mut BRAF mRNA only. siBRAF_WT&Mut (green) was designed to target both WT and Mut BRAF mRNAs. (**B**) Experimental procedure used to investigate the effects of siRNAs against WT and/or Mut BRAF in BxPC-3 cells. (**C**) PCR primers detecting WT and Mut BRAF genes specifically or indiscriminately. Forward and reverse common primers used to detect both WT and Mut BRAF mRNAs simultaneously. The red region indicates the 15-bp deletion in Mut BRAF. The forward primer is common to both WT and Mut BRAF specific primers. The reverse primers are specific to WT and Mut BRAF. Pink indicates the mismatched nucleotide of each primer. (**D**) BRAF mRNA levels quantified by qRT-PCR using total RNA from BxPC-3 cells transfected with 50 nM of each siRNA. At 24 and 48 h after the third transfection, cells were collected and BRAF mRNA levels measured. The common primers detecting both the WT and Mut genes were used in the experiment, and WT and Mut BRAF mRNAs were detected without distinction. The green and blue bars show the results using total RNA purified from the cells collected at 24 and 48 h, respectively. The results indicate the averages of 3 independent experiments. (**E**) WT and Mut BRAF mRNA levels quantified by qRT-PCR using total RNA from BxPC-3 cells transfected with 50 nM of each siRNA. Specific primers were used to measure WT and Mut BRAF mRNA levels separately. siBRAF_WT&Mut and siBRAF_WT significantly suppressed WT BRAF mRNA; siBRAF_WT&Mut, siBRAF_Mut1, and siBRAF_Mut2 significantly suppressed Mut BRAF mRNA. The results indicate the averages of 3 independent experiments. The *p*-values for comparisons of WT and Mut BRAF mRNA levels were determined using Student’s *t*-test (* *p* < 0.05). (**F**) The upper panel indicates the Western blotting using lysates from BxPC-3 cells transfected with siControl or siRNA targeting BRAF. Unphosphorylated ERK1/2 (ERK1/2) and phosphorylated ERK1/2 (P-ERK1/2) were detected using anti-ERK1/2 and anti-phospho-ERK1/2 antibodies, respectively. β-actin was used as a control. All cell samples were collected on day 1 after three transfections with 50 nM siRNA. The lower panel indicates the results of quantitative estimates. The intensity of each band was normalized by intensity of β-actin. (**G**) Cell counts on each day after three transfections. The black, green, red, solid blue, and blue-dotted lines indicate the number of BxPC-3 cells transfected with 50 nM siControl, siBRAF_WT&Mut, siBRAF_WT, siBRAF_Mut1, and siBRAF_Mut2, respectively. The cell numbers were normalized with those on the first day after three transfections. The results indicate the averages of 6 independent experiments.

**Figure 3 cancers-14-03162-f003:**
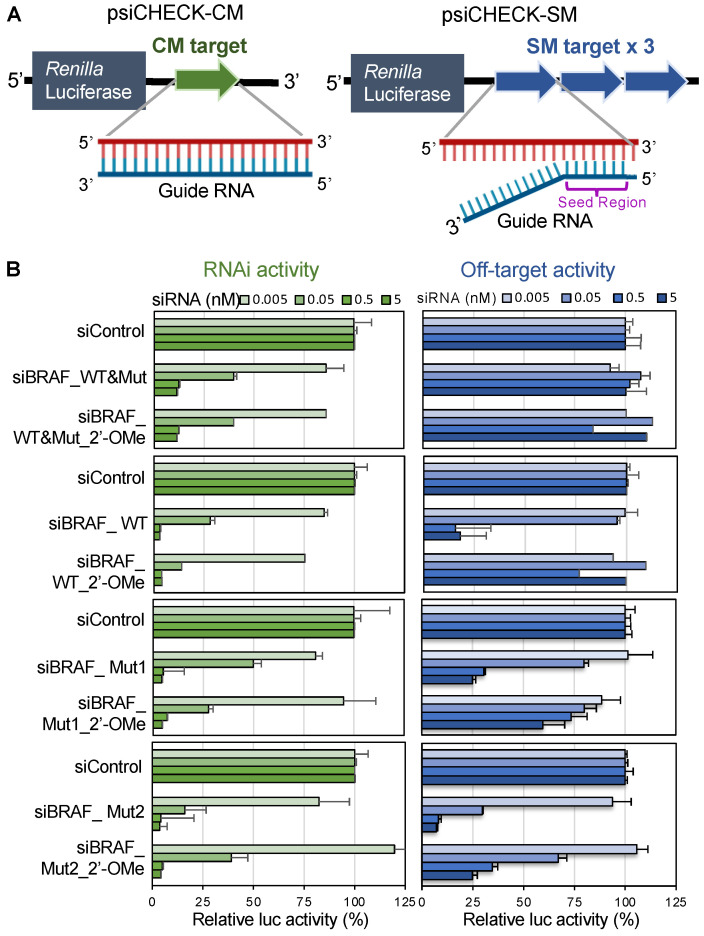
RNAi and off-target activities of each siRNA targeting BRAF in HeLa cells determined by dual luciferase reporter assays. (**A**) The structures of CM (RNAi) or SM (off-target) reporter constructs. (**B**) The CM (RNAi) or SM (off-target) reporter constructs were transfected with an siRNA along with the pGL3-Control firefly luciferase expression construct into HeLa cells, and luciferase activity was measured 1 day after transfection. The RNAi and off-target activities were calculated as *Renilla* luciferase activity normalized to control firefly luciferase activity. RNAi (left, green bars) and off-target activities (right, blue bars) for four siRNA concentrations (0.005, 0.05, 0.5, and 5 nM). The horizontal bars indicate the relative luciferase activity levels. The results indicate the averages of 3 independent experiments.

**Figure 4 cancers-14-03162-f004:**
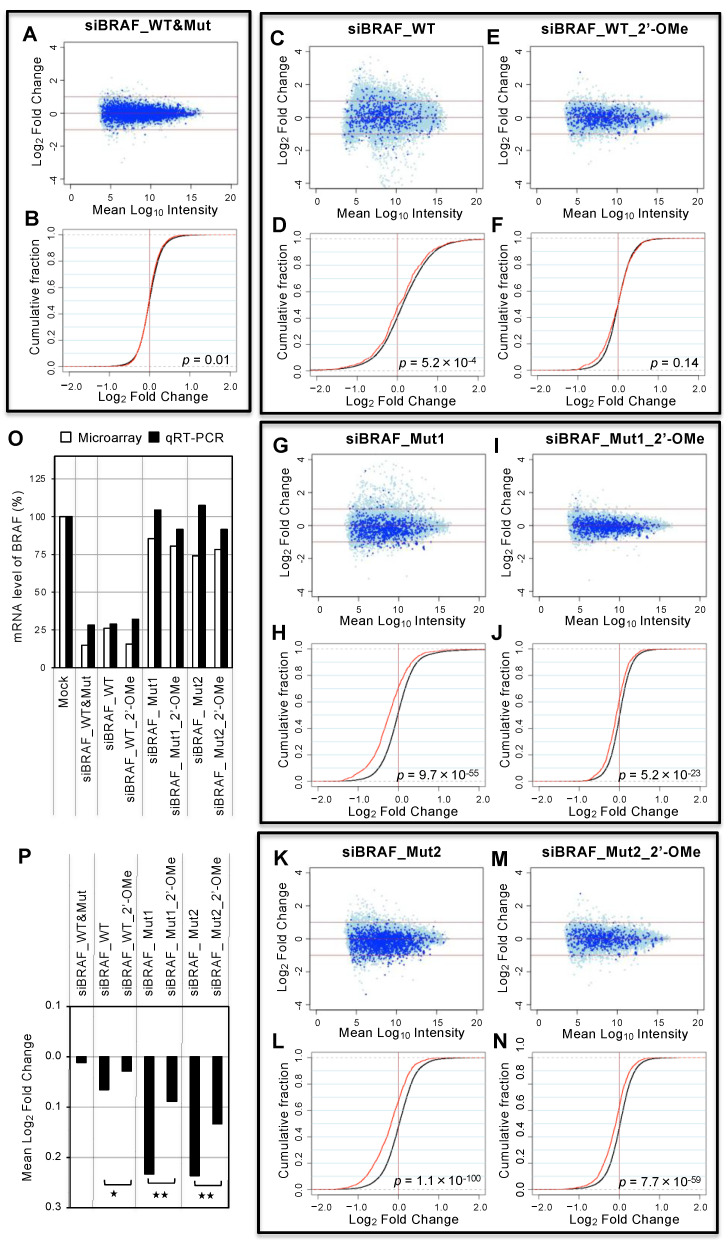
Microarray-based off-target profilings. Microarray profiles of transcripts from the cells transfected with (**A**,**B**) siBRAF-WT&Mut, (**C**,**D**) siBRAF-WT, (**E**,**F**) siBRAF-WT_2′-OMe, (**G**,**H**) siBRAF-Mut1, (**I**,**J**) siBRAF-Mut1_2′-OMe, (**K**,**L**) siBRAF-Mut2, (**M**,**N**) siBRAF-Mut2_2′-OMe. Results are shown in MA plots (**A**,**C**,**E**,**G**,**I**,**K**,**M**) and cumulative frequency distributions (**B**,**D**,**F**,**H**,**J**,**L**,**N**). Blue dots in MA plots indicate the transcripts with SM sequences of each siRNA. The red lines in cumulative frequency distributions indicate the fraction of transcripts with SM sequences in their 3′UTRs. The black lines show the transcripts with no SM sequences. Change in the expression level was determined as the log_2_ of expression in the siRNA-transected cells compared to that in mock-transected control cells. The statistical significance of their dissimilarity was quantified based on the *p*-value using Wilcoxon’s rank-sum test [46]. (**O**) Changes in BRAF expression levels in the cells transfected with each siRNA and determined by microarray and qRT-PCR. (**P**) Mean log2 fold changes of transcripts with SM sequences of each siRNA. The *p*-values were calculated by Wilcoxon’s rank-sum test (* *p* < 0.01, ** *p* < 0.001).

**Figure 5 cancers-14-03162-f005:**
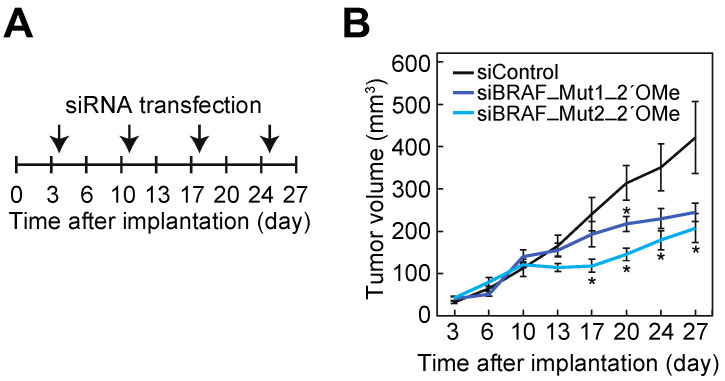
siBRAF suppresses the growth of xenograft tumors. (**A**) Time course of xenograft experiment for the investigation of anti-tumor effects of siBRAF_Mut1_2′-OMe and siBRAF_Mut2_2′-OMe. (**B**) The effect of siBRAF on the tumor growth. Tumor size was measured twice a week, and results are shown as mean with SEM (*n* = 5). The results indicate the averages of 5 independent experiments. The *p*-values were calculated by Student’s *t*-test versus siControl (* *p* < 0.05).

## Data Availability

Microarray data are registered with GEO, accession number GSE189961.

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
