# Peer review of "Knockdown of 15-bp Deletion-Type v-raf Murine Sarcoma Viral Oncogene Homolog B1 mRNA in Pancreatic Ductal Adenocarcinoma Cells Repressed Cell Growth In Vitro and Tumor Volume In Vivo"

_cancers, 2022, doi:10.3390/cancers14133162_

Round 1
Reviewer 1 Report
General comments:
I) Several statements were insufficiently supported by references.
II) Materials and methods lacked information to assess the reproducibility of the experiments by other laboratories.
III) Ethical information was insufficiently delineated.
IV) It was unclear if there were independent measurements to support adequately the findings.
V) The phenotype and functions of the cells were insufficiently monitored.
VI) It was unclear if the transfections maintained the cell function or if they induced a cell differentiation.
VII) The discussion contained one paragraph which shall be moved into the introduction.
Minor comments:
1) Title, p1: Replace BRAF by its full name in the title
2) Simple Summary, lines 13-14, p1: BRAF by its full name in “The BRAF gene containing a 15-base pair (bp) deletion at L485-P490 is a cause of 13 several cancers.”
3) Introduction, from line 44 p1 to line 45 p2: Add reference to support “Mutation of oncogene Kirsten rat sarcoma (KRAS) is the most common cause of pancreatic cancer, accounting for more than 90% of cases. The KRAS gene encodes”
4) Introduction, line 45-46, p2: Add reference to “The KRAS gene encodes a small GTPase transductor protein that binds to guanosine diphosphate (GDP) and guanosine triphosphate (GTP).”
5) Introduction, line 47-48, p2: Add reference to support “KRAS possesses intrinsic GTPase activity, is involved in the Ras/mitogen-activated protein kinase (MAPK) signaling pathway, and plays an important role in cell proliferation.”
6) Introduction, line 49-51, p2: Add reference to “Most KRAS mutations impair the intrinsic activity of GTPase, impede the function of GTPase-accelerating proteins, and inhibit the conversionof GTP to GDP.”
7) Introduction, line 52-53, p2: Add reference “In addition to PDAC harboring KRAS mutations, 10% of PDAC cases harbor wild-type (WT) KRAS.”
8) Introduction, line 53-55, p2: Add reference to support “Among these cases, 40% are related to defects in the activated MAPK pathway caused by oncogenic BRAF mutation, while 20% are due to defective DNA mismatch repair resulting from microsatellite instability in tumors.”
9) Introduction, line 61-62, p2: Add reference to support “Aside from V600E, a 15-base pair (bp) deletion from L485 to P490 in BRAF, an activating mutation, has also been detected.”
10) Introduction, line 61-62, p2: Add reference to support “This 15-bp in-frame deletion has been observed in lung and ovarian cancers in addition to pancreatic cancers.”
11) Introduction, line 64-65, p2: It was unclear why PDAC-derived BxPC-3 cells were selected, add explanations and references to support“In this study, we used PDAC-derived BxPC-3 cells harboring the 15-bp BRAF deletion mutation (Mut) and applied the WT BRAF gene heterogeneity.”
12) Introduction, line 78-79, p2: Add reference to support “Some off-target effects of RNAi have been reported.”
13) Materials and Methods, cells cultures, lines, 106-113,p3: Justify the use of cells, add references to support “Cell culture Human PDAC-derived cell lines, BxPC-3 and AsPC-1, were cultured at 37°C with 5% CO2 in RPMI 1640 medium (Thermo Fisher Scientific) with 10% heat-inactivated fetal bovine serum (FBS, Sigma) and 1% penicillin-streptomycin solution (FUJIFILM Wako). Human HeLa cells, derived from cervical cancer, and OCUB-M cells, derived from breast cancer, were cultured at 37°C with 5% CO2 (DMEM, Thermo Fisher Scientific) with 10% heat-inactivated FBS and 1% penicillin-streptomycin solution.”
14) Materials and Methods, cell cultures, lines, 106-113,p3: Specify how long cells were incubated, how cell phenotype and function were monitored,to support “Cell culture Human PDAC-derived cell lines, BxPC-3 and AsPC-1, were cultured at 37°C with 5% CO2 in RPMI 1640 medium (Thermo Fisher Scientific) with 10% heat-inactivated fetal bovine serum (FBS, Sigma) and 1% penicillin-streptomycin solution (FUJIFILM Wako). Human HeLa cells, derived from cervical cancer, and OCUB-M cells, derived from breast cancer, were cultured at 37°C with 5% CO2 (DMEM, Thermo Fisher Scientific) with 10% heat-inactivated FBS and 1% penicillin-streptomycin solution.”
15) Materials and Methods, Measurements of RNAi and off-target activities by dual luciferase reporter assays, lines 125-126, p3: Justify the use of HeLa cells to support For luciferase reporter assay, HeLa cells were inoculated in a well of 24-well cultureplates (1 × 105 cells/well).
16) Materials and Methods, Measurements of RNAi and off-target activities by dual luciferase reporter assays lines, 125-128, p3: Add more information how transfection was performed to support Measurements of RNAi and off-target activities by dual luciferase reporter assays “Twenty-four hours later, cells were transfected with siRNA (0.005, 0.05, 0.5, or 5nM), 100 ng of pGL3-Control vector (Promega), and 10 ng of the corresponding psiCHECK-1 vector using Lipofectamine 2000 reagent (Thermo Fisher Scientific).”
17) Materials and Methods, Measurements of RNAi and off-target activities by dual luciferase reporter assays lines 130-131, p3: It was unclear if cells were functional to support “Control siRNA, siControl, does not target neither CM- nor SM-reporter constructs. At 24 hours after transfection, cells were lysed by 1 × passive lysisbuffer (Promega).”
18) Materials and Methods, Quantitative reverse transcription-polymerase chain reaction (qRT-PCR), Title, line 138, p3: Delete qRT-PCR in the title.
19) Materials and Methods, Quantitative reverse transcription-polymerase chain reaction (qRT-PCR),lines 138-140, p3: Justify the use of BxPC-3 and AsPC-1 cell in “BxPC-3 and AsPC-1 cell suspensions (0.5 × 105 cells / well) were seeded into each well of 24-well culture plate 24 hours before transfection, respectively.”
20) Materials and Methods, Quantitative reverse transcription-polymerase chain reaction (qRT-PCR),lines 140-143, p3: Add more information how transfections were performed in “Cells were transfected 140 with 50 nM siRNA by Lipofectamine 2000 or Lipofectamine RNAiMAX reagent (Thermo Fisher Scientific) in 3 consecutive days. At 24 or 48 hours after 3rd times siRNA transfec tion, the cells were collected”
21) Materials and Methods, Quantitative reverse transcription-polymerase chain reaction (qRT-PCR),lines 140-143, p3: Add more information how RNAs were extracted in “Total RNAs were extracted from the cells using the FastGene RNA Premium Kit (Nippon Genetics).”
22) Materials and Methods, Quantitative reverse transcription-polymerase chain reaction (qRT-PCR),lines 147-148, p4: Add more information how qRT-PCR was performed to support “qRT-PCR was pe formed using the KAPA SYBR Fast qPCR Kit (KAPABIOSYSTEMS) with the StepOnePlus Real-…”
23) Materials and Methods, Western blotting, lines 154-156, p4: Specify, DTT,NP-40 and protease inhibitor in “The cells were washed with PBS and lysed in lysis buffer (10 mM Hepes-NaOH (pH 7.9), 1.5 mM MgCl2, 10 mM KCl, 0.5 mM DTT, 140 mM NaCl, 1 mM EDTA, 1 mM Na3VO4, 10 mM NaF, 0.5% NP-40 and complete protease inhibitor).”
24) Materials and Methods, Western blotting, lines 156-160, p4: Delete (SDS-PAGE) and (PVDF) since they were never cited in the text in “After determining the protein concentration and denaturization, the samples were separated by sodium dodecyl sulfate polyacrylamide gel electrophoresis (SDS-PAGE), and transferred to a Immobilon- P polyvinylidene fluoride (PVDF) membrane (Merck Millipore) using the Mini Trans-Blot Cell (Bio-Rad).”
25) Materials and Methods, Western blotting, lines 160-163, p4: Specify TBS-T in “The membrane was blocked for 1 hour in Tris-buffered saline with Tween 20 (TBS-T; 20 mM TrisHCl [pH 7.5], 150 mM NaCl, 0.1% Tween) supplemented with 5% skim milk (FUJIFILM Wako) at room temperature, and incubated with specific first antibodies in TBS-T with 5% skim milk at 4°C overnight.”
26) Materials and Methods, Western blotting, lines 167-169, p4: Specify ERK1/2 in “The used first antibodies are as follows; rabbit anti-phospho-ERK1/2 antibody at 1: 1000 dilution (Cell Signaling), rabbit anti-ERK1/2 antibody at 1:1000 dilution (Cell Signaling), and mouse anti-actin antibody at 1: 4000 dilution (Sigma).”
27) Materials and Methods, Western blotting, lines 169-171, p4: Specify HRP in “The used second antibodies are as follows: HRP-linked anti-rabbit or anti-mouse antibody at 1:3000 dilution (GE Healthcare).”
28) Materials and Methods, Microarray, lines 174-175, p4: Add more information how cells were transfected, specify cellsand how cells were monitored in “Cells were transfected three times with 50 nM of eachsiRNA using Lipofectamine 2000.”
29) Materials and Methods, Microarray, lines 175-178, p4: Add more information how total RNA was extracted and specify cells in “Total RNA of the harvested cells was purified using FastGene RNA Premium Kit (NIPPON Genetics, Tokyo, Japan), and the RNA quality was confirmed using NanoDrop 2000 spectrophotometer (Thermo Fisher Scientific) and Bioanalyzer (Agilent Technologies, Santa Clara, CA, USA).”
30) Materials and Methods, Microarray, lines 187-189, p4: Specify MA in “Results were shown in MA plots and cumulative accumulations. Furthermore, the aliguot of total RNA was used for qRT-PCR for validation of microarray data.”
31) Materials and Methods, Xenograft mouse model, lines 198-200, p5: Add more information how tumor size was measured to support “Tumor size was measured twice a week and the tumor volume was calculated using the following formula: V = L x S2 (L: longer diameter of tumor, S: shorter diameter of tumor).”
32) Materials and Methods, Xenograft mouse model, lines 200-201, p5: Add specific information to support “All the animal experiments were approved by the Animal Ethics Committee of the University of Tokyo.”
33) Results, Effect of KRAS knockdown in PDAC-derived cell lines AsPC-1 and BxPC-3, with or without Mut KRAS, Title, lines 212-213, p5: Replace KRAS by its full name in the title.
34) Results, Effect of KRAS knockdown in PDAC-derived cell lines AsPC-1 and BxPC-3, with or without Mut KRAS,lines 216-217, p5: Add references to support “AsPC-1 cells harbor homozygous KRAS mutations in the GTP binding sites (35G>A) and WT BRAF, whereas BxPC-3 cells harbor WT KRAS gene and Mut BRAF (15-bp deletion at L485-P490).”
35) Results, Effect of KRAS knockdown in PDAC-derived cell lines AsPC-1 and BxPC-3, with or without Mut KRAS,lines 218-219, p5: It was unclear, rephrase “The siKRAS used in this study repressed the expression ofboth WT and 35G>A-mutated KRAS genes.”
36) Results, Effect of KRAS knockdown in PDAC-derived cell lines AsPC-1 and BxPC-3, with or without Mut KRAS,lines 221-223, p5: Indicated number of independent measurements in Fig-1B legend to support “Relative to the control cells transfected with the control siRNA (siControl), the KRAS mRNA level was effectively decreased to 21.2% in siKRAS-treated AsPC-1 cells (Figure 1B).”
37) Results, Effect of KRAS knockdown in PDAC-derived cell lines AsPC-1 and BxPC-3, with or without Mut KRAS, lines 223-225, p5: Specify ERK and MAPK in “One of the major downstream effects of KRAS activation is phosphorylation of the rapidly accelerated fibrosarcoma MAPK kinase ERK [34,35].”
38) Results, Effect of KRAS knockdown in PDAC-derived cell lines AsPC-1 and BxPC-3, with or without Mut KRAS, lines 225-228, p5: It was unclear if cells remained functional in “Therefore, to investigate the effect of siKRAS on this downstream phosphorylation, western blotting was performed to quantify the phosphorylation levels of ERK1 and ERK2. siKRAS effectively suppressed the phosphorylation of ERK1/2 in siKRAS-transfected AsPC-1 cells (Figure 1C).”
39) Results, Effect of KRAS knockdown in PDAC-derived cell lines AsPC-1 and BxPC-3, with or without Mut KRAS, lines 228-230, p5: Indicated number of independent measurements in Fig-1D legend to support “Next, the effect of siKRAS on cell growth was measured. The cell numbers relative to those on day 229 1 after the last siKRAS transfection were determined on days 2, 3, 4, and 5 (Figure 1D).”
40) Results, Effect of KRAS knockdown in PDAC-derived cell lines AsPC-1 and BxPC-3, with or without Mut KRAS, lines 232-235, p5: Indicated number of independent measurements in Fig-1E and Fig-1G legend sto support “The same experiments were performed in BxPC-3 cells. Although siKRAS significantly decreased the KRAS mRNA level to approximately 6% of the control level (Figure 1E), it showed negligible effects on the phosphorylation levels 234 of ERK1/2 (Figure 1F) and the number of BxPC-3 cells (Figure 1G).”
41) )Results, Effect of KRAS knockdown in PDAC-derived cell lines AsPC-1 and BxPC-3, with or without Mut KRAS, lines 235-237, p5: It was unclear if cells were functunial after the transfection to support “Thus, suppression of he KRAS expression by siKRAS in AsPC-3 cells successfully repressed KRAS mRNA expression, KRAS downstream signaling, and cell number.”
42) Results, Effect of knockdown of Mut BRAF in BxPC-3 cells, line, 265-266, p6: It was unclear if BxPC-3 cells were functional after their transfection to support “Each siRNA (50 nM) was transfected into BxPC-3 cells daily on days 13 using Lipofectamine 2000 (Figure 2B).”
43) Results, Effect of knockdown of Mut BRAF in BxPC-3 cells, line, 271-272, p6: Indicate number of independent measurements in Figure-2D legend to support “At 24 hours after transfection, siBRAF_WT was as effective as siBRAF_WT&Mut and sufficiently decreased the BRAF mRNA level (Figure 2D).”
44) Results, Effect of knockdown of Mut BRAF in BxPC-3 cells, line, 282-285, p7: Indicate number of independent measurements in Figure-2E legend to support “Using these primers, it was revealed that both siBRAF_WT&Mut and siBRAF_WT effectively suppressed the expression of WT BRAF mRNA (Figure 2E), and siBRAF_WT&Mut, siBRAF_Mut1, and siBRAF_Mut2 effectively suppressed the expression of Mut BRAF mRNA (Figure 2E). ”
45) Results, Effect of knockdown of Mut BRAF in BxPC-3 cells, line, 315-318, p8: Provide quantitative estimates of Western Blot to support “Western blotting revealed nearly the same ERK1/2 phosphorylation level between siBRAF_WT and siControl transfections, whereas siBRAF_WT&Mut, siBRAF_Mut1, and siBRAF_Mut2 markedly suppressed ERK1/2 phosphorylation (Figure 2F).”
46) Results, Effect of knockdown of Mut BRAF in BxPC-3 cells, line, 319-320, p8: Indicate number of independent measurements in Figure-2G legend to support “The cell numbers relative to those on day 1 after siRNA transfection were determined on days 2, 3, 4, and 5 (Figure 2G).”
47) Results, Effects of BRAF siRNAs on RNAi and off-target activities, title, line 329, p8: Replace BRAF by its full name in the title.
48) Results, Effects of BRAF siRNAs on RNAi and off-target activities,l ines 336-338, p8: It was unclear if HeLa cells were functional and if they keept their phenotype after their transfection to support “psiCHECK-CM and psiCHECK-SM were individually transfected into human HeLa cells along with pGL3-Control, which expresses the firefly luciferase gene as a control, and siRNAs at various concentrations.”
49) Results, Effects of BRAF siRNAs on RNAi and off-target activities, from line 338, p8 to line 342, p9: Indicate number of independent measurements in Figure-3B legend to support “Luciferase activity was measured 1 day after 3 siBRAF_WT&Mut strongly suppressed Renilla luciferase activity in the cells n harboring the CM target sequence in the of the luciferase gene, but not in the cells harboring the SM target sequence (Figure 3B), indicating that siBRAFWT&Mut has strong RNAi activity but negligible off-target activity.”
50) Results, Effects of BRAF siRNA with 2’OMe modification in the seed region on RNAi and off-target activities, title lines 352-353, p9: Replace BRAF by its full name in the title.
51) Results, Effects of BRAF siRNA with 2’OMe modification in the seed region on RNAi and off-target activities, lines 358-360, p9: It was unclear if modified siRNAs affected functions of the cells to support “The 2’-OMe modified siRNAs were designated as siBRAF-2’OMe, siBRAF_WT_2’OMe, siBRAF_Mut1-2’OMe, and siBRAF_Mut2_2’OMe, respectively. Then, luciferase reporter assays were performed using these modified siRNAs (Figure 3B).”
52) Results, Genome-wide analyses of siRNA off-target effects by microarray, lines 378-385, p9: Indicate reproducibility of Fig 4 and specify SM to support “The MA plots (M = intensity ratio, A = average intensity) of the microarray data using HeLa cells (Figure 4A, C, E, G, I, K, M) indicated changes in the expression levels of the transcripts, and dark blue circles indicate the off-target transcripts with SM sequence(s) in their 3 ́ UTRs. The cumulative distribution indicated the averaged fold-changes of the off-target transcripts compared to those with non-SM transcripts (Figure 4B, D, F, H, J, L, N). The expression levels of the WT BRAF gene in HeLa cells were significantly downregulated by siBRAF_WT and siBRAF_WT&Mut, but not by siBRAF_Mut1 and siBRAF-Mut2 (Figure 4O).”
53) Results, Suppression of cell growth of BxPC-3 cells by siBRAF-Muts in mice, title line 424, p12: Replace BRAF by its full name in the tile.
54) Results, Suppression of cell growth of BxPC-3 cells by siBRAF-Muts in mice, lines 426-433, p12: It was unclear if the transfected cells were functional and if they kept their wild-type phenotype to support “At first, each siRNA (50 nM) was transfected into BxPC-3 cells daily on days 13 using Lipofectamine 2000. At 24 hours after the third transfection, cells were collected and qRT-PCR was performed to quantify the expression of WT and Mut BRAF mRNAs, respectively, using each of distinguishable PCR primers (Figure 2C) for WT or Mut BRAF mRNAs. All of siBRAF siRNAs (siBRAF-WT, siBRAF-Mut1, siBRAF-Mut2) with 2 ́-OMe modifications showed the similar levels of suppression on BRAF mRNAs compared to the not-modified ones (Supplementary Figure S4).”
55) Results, Suppression of cell growth of BxPC-3 cells by siBRAF-Muts in mice, lines 436-437, p12: Indicate number of independent measurements to support statistitics in the Figure-5B legend in “The growth of tumor was suppressed by the treatment with siBRAF_Mut2’OMe or siBRAF-2’-OMe, compared to the result of siControl treatment (Figure 5B).”
56) Discussion, lines 446-449, p12: Delete “In this study, we developed siRNAs which specifically knock down Mut BRAF 446 mRNA harboring 15-bp deletion but not affect WT BRAF expression. The BxPC-3 cell line 447 was used to analyze the effect of these siRNAs, as this cell line has no KRAS mutation and 448 is heterozygous for the 15-bp deletion of BRAF.”
57) Discussion, lines,449-451, p12: Move into the introduction “Structurally, the region of BRAF containing 15-bp deletion is involved in the beta3/alphaC helix loop, and this deletion shortens the beta3/alphaC homologous loop within the kinase domain [37]. Therefore, the 15-bp deletion reduces flexibility by locking the helix into the active C-alpha helix conformation, generating a onstitutively active form [35].”
58) Discussion, lines 453-454, p12: Add explanation, references and move into the introduction “Thus, this deletion is considered to be a cause of carcinogenesis.”
59) Discussion, lines 456-460 and lines 464-468, p13: It was unclear if the silencing maintained the cell phenotype or induced cell differentiation. Rephrase “In contrast, siKRAS suppressed the growth of AsPC-1 cells, which are homozygous for the oncogenic KRAS mutation. Therefore, the results suggest that the 15-bp deletion of BRAF mutation functions as the main cause of carcinogenesis independent of KRAS expression in BxPC-1 cells.” and in “s the siRNA targeting WT BRAF (siBRAF_WT) did not suppress ERK1/2 phosphorylation or cell growth, the approximately 70% reduction in WT BRAF mRNA expression appears to have a negligible effect on cell growth (Figure 2E and G). In contrast, the approximately 70% reduction in Mut BRAF mRNA might be sufficient to suppress down-stream carcinogenic signaling.”
60) Discussion, lines 493-494, p13: Add reference to support “Even though siRNA is a powerful tool for gene silencing, many challenges and barriers to 493 siRNA therapeutics remained to be overcome.”
Reviewer 2 Report
I overall enjoyed the manuscript by the authors. SiRNA is a promising therapeutic modality for various disease, including cancer. I would ask the authors to comment or provide additional data on the following. The in vivo tumor model demonstrates that the tumors in mice still grew (albeit slower) after siRNA knockout. If BRAF mutation is thought to be the causative mutation causing this melenoma, what other mechanisms/pathways continue to cause proliferation of tumor cells?
Reviewer 3 Report
The current study describes the use of siRNA-mediated knockdown of specifically the mutated version of a gene (BRAF). The authors also address the off-target effects of siRNA-mediated treatments and provide evidence for overcoming these effects. Overall, the study is well designed, and the comments below are provided to understand some points better.
1. Introduction: Could the authors include the incidence rate of the 15bp deletion mutation observed in the BRAF gene?
2. Could the authors add a few sentences about the need for selective reduction of the mutant BRAF protein expression? What is the significance of this mutation, and then talk about the siRNA approach for the same?
3. Results Section 3.2: It would be beneficial for the readers if the authors could provide a small rationale regarding the use of siKRAS and what question they intend to address with the experiment.
4. Could the authors explain why a cell line with wild KRAS and BRAF was not used as a control for these experiments?
5. Section 3.3 Could the authors explain the rationale for switching the cell lines to Hela and OCUB-M for these experiments?
6. Section 3.4 Could the authors comment on the effect of this modification on the expression of BRAF in the BxPC-3 cell lines?
7. The discussion (line 488) mentions no effect observed on the innate immune response. Could the authors please comment on the reduction observed upon a comparison of modified to unmodified siRNA
8. Also, the article's title mentions repression of cell growth in response to siRNA treatment. While this may be true for the in vitro experiments, the in vivo experiments show a reduction in tumor volumes. Maybe the title could be modified to reflect the same.
Round 2
Reviewer 1 Report
Major comments:
Most of the comments were adequately addressed, except:
V) The phenotype and functions of the cells were insufficiently monitored
Minor comments:
14) Materials and Methods, cell cultures, lines, 106-113,p3: Specify how long cells were incubated, how cell phenotype and function were monitored,to support “Cell culture Human PDAC-derived cell lines, BxPC-3 and AsPC-1, were cultured at 37°C with 5% CO2 in RPMI 1640 medium (Thermo Fisher Scientific) with 10% heat-inactivated fetal bovine serum (FBS, Sigma) and 1% penicillin-streptomycin solution (FUJIFILM Wako). Human HeLa cells, derived from cervical cancer, and OCUB-M cells, derived from breast cancer, were cultured at 37°C with 5% CO2 (DMEM, Thermo Fisher Scientific) with 10% heat-inactivated FBS and 1% penicillin-streptomycin solution.”
→We did not check the cell phenotype or functions systematically. However, we think the morphologies of these cells are not changed. Therefore, we added the cell number registered in each cell bank from which the cells were obtained (p.3, lines 167-168, and 172), and described in the acknowledgments (p.15, lines 711-714).
Reviewer’s answer: It was insufficiently addressed. Cell phenotype and functions shall be checked.
15) Materials and Methods, Measurements of RNAi and off-target activities by dual luciferase reporter assays, lines 125-126, p3: Justify the use of HeLa cells to support For luciferase reporter assay, HeLa cells were inoculated in a well of 24-well culture plates (1 × 105 cells/well).
→We are sorry, we used three types of cells, HeLa, AsPC-1, and OCUB-1 cells, since we think that the similar results are obtained by any cells for reporter assays. Therefore, we added the AsPC-1 and OCUB-1 cells in the text (p.3, lines 187-188).
Reviewer’s answer: It was unclear why these cells were selected. Add justifications of the use of these three types of cells.
17) Materials and Methods, Measurements of RNAi and off-target activities by dual luciferase reporter assays lines 130-131, p3: It was unclear if cells were functional to support “Control siRNA, siControl, does not target neither CM- nor SM-reporter constructs. At 24 hours after transfection, cells were lysed by 1 × passive lysisbuffer (Promega).”
→We added the references [32,37] to support this sentence indicating that siControl does not target neither CM- nor SM-reporter constructs. In these references, it was shown that the cells showed no apparent abnormalities to measure RNAi and off-target activities (p.4, line198).
Reviewer’s answer: Apparently the cell functions were not monitored. Yes, there were previous reports that indicated no changes during the transfections. However it is a good practice to check if it was correct.
23) Materials and Methods, Western blotting, lines 154-156, p4: Specify, DTT,NP-40 and protease inhibitor in “The cells were washed with PBS and lysed in lysis buffer (10 mM Hepes-NaOH (pH 7.9), 1.5 mM MgCl2, 10 mM KCl, 0.5 mM DTT, 140 mM NaCl, 1 mM EDTA, 1 mM Na3VO4, 10 mM NaF, 0.5% NP-40 and complete protease inhibitor).”
→ The companies of DTT, NP-40, and complete protease inhibitor were specified.
Reviewer’s answer: Specify the abbreviations DTT (is that dithiothreitol?) and NP-40 (=?) since the abbreviations appeared for the first time.
26) Materials and Methods, Western blotting, lines 167-169, p4: Specify ERK1/2 in “The used first antibodies are as follows; rabbit anti-phospho-ERK1/2 antibody at 1: 1000 dilution (Cell Signaling), rabbit anti-ERK1/2 antibody at 1:1000 dilution (Cell Signaling), and mouse anti-actin antibody at 1: 4000 dilution (Sigma).”
→The companies and product numbers of Anti-phospho-ERK1/2 antibody, rabbit anti-ERK1/2 antibody, and anti-actin antibody were specified (p.4, lines 238, 240, 241, and 242).
Reviewer’s answer: Specify ERK1/2 abbreviation as “extracellular signal-regulated kinase 1/2” since the abbreviation appeared for the first time.
27) Materials and Methods, Western blotting, lines 169-171, p4: Specify HRP in “The used second antibodies are as follows: HRP-linked anti-rabbit or anti-mouse antibody at 1:3000 dilution (GE Healthcare).”
→The company and product number of HRP-linked anti-rabbit or anti-mouse antibody was specified (p.4, lines 243-244).
Reviewer’s answer: Specify HRP Horseradish peroxidase since the abbreviation appeared for the first time.
28) Materials and Methods, Microarray, lines 174-175, p4: Add more information how cells were transfected, specify cells and how cells were monitored in “Cells were transfected three times with 50 nM of each siRNA using Lipofectamine 2000.”
→We added that the transfected cells were HeLa cells, transfection was performed according to the manufacture’s protocol (p.5, lines 253-255). The cells were not monitored, but the morphologies of the cells were not changed.
Reviewer’s answer: It was insufficiently addressed.
37) Results, Effect of KRAS knockdown in PDAC-derived cell lines AsPC-1 and BxPC-3, with or without Mut KRAS, lines 223-225, p5: Specify ERK and MAPK in “One of the major downstream effects of KRAS activation is phosphorylation of the rapidly accelerated fibrosarcoma MAPK kinase ERK [34,35].”
→We specified ERK as ERK1/2, and changed this sentence as “One of the major downstream effects of KRAS activation is phosphorylation of the rapidly accelerated fibrosarcoma MAPK kinases ERK1/2 [12,42].” (p.6, line 326)
Reviewer’s answer: It was insufficiently addressed. MAPK shall be specified as “Mitogen-activated protein kinase” since the abbreviation appeared for the first time.
38) Results, Effect of KRAS knockdown in PDAC-derived cell lines AsPC-1 and BxPC-3, with or without Mut KRAS, lines 225-228, p5: It was unclear if cells remained functional in “Therefore, to investigate the effect of siKRAS on this downstream phosphorylation, western blotting was performed to quantify the phosphorylation levels of ERK1 and ERK2. siKRAS effectively suppressed the phosphorylation of ERK1/2 in siKRAS-transfected AsPC-1 cells (Figure 1C).”
→To indicate that the cells remained functional, we changed this sentence to suggest that the downstream signalling pathway is active, as “Therefore, to investigate the effect of siKRAS on this downstream phosphorylation, western blotting was performed to quantify the phosphorylation levels of ERK1 and ERK2. siKRAS effectively suppressed the phosphorylation of ERK1/2 in siKRAS-transfected AsPC-1 cells (Figure 1C), suggesting that the cells are active to response to KRAS knockdown which suppresses the downstream cell proliferation signalling. Therefore,---” (p.6, lines 326-331)
Reviewer’s answer: It was insufficiently addressed.
41) )Results, Effect of KRAS knockdown in PDAC-derived cell lines AsPC-1 and BxPC-3, with or without Mut KRAS, lines 235-237, p5: It was unclear if cells were functunial after the transfection to support “Thus, suppression of he KRAS expression by siKRAS in AsPC-3 cells successfully repressed KRAS mRNA expression, KRAS downstream signaling, and cell number.”
→To show the cells are functional, we added the statement “, suggesting that the AsPC-3 cells are functional to response to the KRAS knockdown” at the end of this sentence (p.6, lines 340-341).
Reviewer’s answer: It was insufficiently addressed.
42) Results, Effect of knockdown of Mut BRAF in BxPC-3 cells, line, 265-266, p6: It was unclear if BxPC-3 cells were functional after their transfection to support “Each siRNA (50 nM) was transfected into BxPC-3 cells daily on days 13 using Lipofectamine 2000 (Figure 2B).”
→We added the sentence, “These transfections were considered to show the fatal damages to the cells, since the downstream responses by BxPC-3 knockdown were successfully observed.” to suggest the cells are functional (p.7, lines 378-380).
Reviewer’s answer: It was insufficiently addressed.
48) Results, Effects of BRAF siRNAs on RNAi and off-target activities,l ines 336-338, p8: It was unclear if HeLa cells were functional and if they keept their phenotype after their transfection to support “psiCHECK-CM and psiCHECK-SM were individually transfected into human HeLa cells along with pGL3-Control, which expresses the firefly luciferase gene as a control, and siRNAs at various concentrations.”
→We added the references [32,37], which are our previous papers to support this procedure is functional, in the first sentence of this paragraph (p.9, lines 450-451).
Reviewer’s answer: It was insufficiently addressed.
51) Results, Effects of BRAF siRNA with 2’OMe modification in the seed region on RNAi and off-target activities, lines 358-360, p9: It was unclear if modified siRNAs affected functions of the cells to support “The 2’-OMe modified siRNAs were designated as siBRAF-2’OMe, siBRAF_WT_2’OMe, siBRAF_Mut1-2’OMe, and siBRAF_Mut2_2’OMe, respectively. Then, luciferase reporter assays were performed using these modified siRNAs (Figure 3B).”
→We added the reference [37,43], which is our previous paper to support this procedure is functional (p.10, line 477).
Reviewer’s answer: It was insufficiently addressed.
Reviewer 3 Report
The authors have addressed the comments satisfactorily
Author Response
Thank you very much for the reviewer's comment, "The authors have addressed the comments satisfactorily".
Round 3
Reviewer 1 Report
The authors addressed adequately most of my concerns.
I do not have further comments.
This manuscript is a resubmission of an earlier submission. The following is a list of the peer review reports and author responses from that submission.
Round 1
Reviewer 1 Report
In 2018, FDA approved first siRNA drug patisiran. siRNA therapy therefore has set up an new milestone, as it has changed the treatment of human diseases. In this manuscript Song et al. design and test siRNAs specifically targeting 15-bp deletion-type Mut BRAF with no or little effect on the expression of wild type (WT) BRAF gene. The 15-bp deletion locks the conformation of BRAF to be constitutively active and is considered as carcinogenesis. The 15-del specific siRNAs effectively knockdown Mut BRAF mRNA expressions more than 50%, but not affect WT BRAF. The Mut BRAF specific knockdown markedly suppresses ERK1/2 phosphorylation and represses the growth of pancreatic ductal adenocarcinoma cells. Moreover, 2’-O-methy modification dramatically improves the seed-dependent off target effect of siRNAs, which is examined by luciferase reporter assay. Furthermore, genome-wide analyses of siRNA off-target effects are investigated by microarray. Similarly, 2’-Ome modification clearly suppresses global off-target effects.
The manuscript is well written, and data are nicely presented. The off-target effects are objectively evaluated, which is critical for siRNA development and its therapeutic application.
Even though the novelty of the study is limited since nothing is new in terms of siRNA design and modification, the study is of significance because it is the first approach targeting the 15 bp deletion of BRAF gene, one of the important causes of pancreatic ductal adenocarcinoma.
While this reviewer supports the eventual acceptance of this manuscript, several suggestions are given as below. In vivo animal study is encouraged to strengthen the clinical significance of the study:
- The study shows the knockdown of 15 bp deletion type BRAF mRNA markedly suppresses ERK1/2 phosphorylation and represses cell growth. In comparison, siRNAs targeting WT neither reduce ERK1/2 phosphorylation nor affect cell growth. These data suggest siRNAs targeting 15 bp deletion affect cell growth through ERK1/2 phosphorylation. However to further confirm this point, a constitutively phosphorylated ERK1/2 mutant can be expressed to observe whether it abolishes the repression on cell growth of pancreatic ductal adenocarcinoma.
- The in vitro data are nice and exciting. However, a major obstacle to the therapeutic application of siRNAs is that siRNAs may stimulate the innate immune system and trigger produce unwanted toxicities. Also, what delivery system to choose and any further modifications are needed to protect siRNAs degradation? Therefore, an in vivo testing of the siRNAs in animal model would be appreciated to validate its clinical feasibility and significance.
Reviewer 2 Report
Hello,
This manuscript deciphering "Knockdown of 15-bp Deletion-type BRAF mRNA Represses Growth of Pancreatic Ductal Adenocarcinoma Cells" is focusing on an interesting aspect of BRAF mutation in a deadly cancer. As authors addressed the lack of drugs for targeting mutant BRAF drags interest in this field. The authors chose the siRNA mediated approach to target the mutant BRAF which reduced the growth of BXPC3 cells. The manuscript has following aspects:
Strengths:
1) Methods were well defined.
2) Experiments had biological replicates.
3) Figures were well presented with excellent statistics.
Weaknesses:
1) Using a single cell line BXPC3 may not justify the concept of inhibiting PDAC using the specific siRNA!
2) In vitro models are not complete validation for the current hypothesis. In vivo study is strongly recommended.
3) HeLa cells might be helpful to your study. In this case another PDAC cell lien is required as a minimum model to support the hypothesis.
4) Most of the PDAC cells revert to different drugs. How about the growth of the cells in long term survival assays?
5) Authors stated about the siRNA stability in discussion. In that case how will the current siRNA remain stable in appropriate models. There is no validation.
Overall, the manuscript is mostly development of a targeting siRNA and validation in minimal models which needs more experimental models to establish it as an effective agent.
Thank You!
Round 2
Reviewer 1 Report
As I mentioned in my original review comments,
As I mentioned in the my original review comments, the manuscript is extremely well written, and data are nicely presented. I read carefully including supplementary documents, and find it's almost flawless. But its novelty is limited since nothing is new in terms of siRNA design and modification. If the authors could add animal experiment to prove the in vivo efficacy as well as toxicity, the significance of the study is highlighted.
However, the authors couldn't provide in vivo experiment results at this point. I would think the overall quality of the study hasn't met the standards of Cancers journal. I don't recommend to accept the manuscript at this point.